# The early-acting glycosome biogenic protein Pex3 is essential for trypanosome viability

Hiren Banerjee, Barbara Knoblach, Richard A Rachubinski

**Trypanosomatid parasites are infectious agents for diseases such as African sleeping sickness, Chagas disease, and leishmaniasis that threaten millions of people, mostly in the emerging world. Trypanosomes compartmentalize glycolytic enzymes to an organelle called the glycosome, a specialized peroxisome. Functionally intact glycosomes are essential for trypanosomatid viability, making glycosomal proteins as potential drug targets against trypanosomatid diseases. Peroxins (Pex), of which Pex3 is the master regulator, control glycosome biogenesis. Although Pex3 has been found throughout the eukaryota, its identity has remained stubbornly elusive in trypanosomes. We used bioinformatics predictive of protein secondary structure to identify trypanosomal Pex3. Microscopic and biochemical analyses showed trypanosomal Pex3 to be glycosomal. Interaction of Pex3 with the peroxisomal membrane protein receptor Pex19 observed for other eukaryotes is replicated by trypanosomal Pex3 and Pex19. Depletion of Pex3 leads to mislocalization of glycosomal proteins to the cytosol, reduced glycosome numbers, and trypanosomatid death. Our findings are consistent with *Pex3* being an essential gene in trypanosomes.**

## Introduction

Neglected tropical diseases form a group of infectious parasitic diseases that affect a large percentage of the world population, mostly in emerging nations (Stuart et al, 2008; Mitra & Mawson, 2017; World Health Organization, 2017). Sleeping sickness affects millions in sub-Saharan Africa and is caused by the protozoan parasite, *Trypanosoma brucei*, which is transmitted by the tsetse fly. If left untreated, sleeping sickness is fatal. Animal trypanosomiases caused by other members of the genus *Trypanosoma* place additional financial and social pressures on the nations of this region. Current drugs used to treat sleeping sickness are restricted in their utility because of toxicity, severe side effects, and complicated administration (Barrett & Croft, 2012; Giordani et al, 2016; de Rycker et al, 2018). Moreover, neglected tropical diseases affect mostly those countries with the least financial or infrastructural resources to develop or deliver new therapies. Therefore, the identification of novel drug targets and the development of new drugs for these targets are pressing priorities.

Trypanosomatid parasites contain a specialized peroxisome called the glycosome that compartmentalizes enzymes of the glycolytic pathway (Haanstra et al, 2016), which are located in the cytosol of cells of other organisms. Because of the high rate of glycolysis carried out by trypanosomatids, a failure to sequester glycolytic enzymes away from the cytosol results in uncontrolled glucose phosphorylation, accumulation of glucose metabolites to toxic levels, depletion of ATP stores, and ultimately death of the parasites (Furuya et al, 2002). Because glycosomes are both unique to trypanosomatids and essential for their viability, they pose an ideal target for drug development.

The Pex3 protein has been ascribed the role of master regulator of peroxisome biogenesis in all organisms studied, including fungi, plants, and animals. Pex3 functions in the de novo biogenesis of peroxisomes via its interactions with the peroxisomal membrane protein receptor, Pex19 (Ghaedi et al, 2000; Fang et al, 2004), and the peroxisomal membrane protein, Pex16 (Matsuzaki & Fujiki, 2008). When Pex3 is absent, cells lack peroxisomes entirely (Smith & Aitchison, 2013; Farré et al, 2018), but reintroduction of the *Pex3* gene re-establishes peroxisome biogenesis and the peroxisome complement of a cell (Hoepfner et al, 2005). Although Pex3 has been found throughout the diversity of the eukaryota, its identity in trypanosomatids has remained elusive. Pex19 (Banerjee et al, 2005) and Pex16 (Kalel et al, 2015) have been identified in trypanosomes, and the specificity of peroxisomal membrane protein binding to Pex19 has been conserved between trypanosomes and mammals (Saveria et al, 2007), thus making it likely that glycosomes follow a de novo biogenic pathway similar to that of peroxisomes in other organisms (Bauer & Morris, 2017). As RNAi knockdown of the *Pex16* or *Pex19* gene in trypanosomes was not lethal (Banerjee et al, 2005; Kalel et al, 2015), identification of the *Pex3* gene in trypanosomatids has continued to be of paramount importance.

---

Department of Cell Biology, University of Alberta, Edmonton, Canada

Correspondence: rick.rachubinski@ualberta.ca

## Results and Discussion

### A putative trypanosomal Pex3 is identified by protein secondary structure predictive bioinformatics

Prediction of a trypanosomatid Pex3 by primary structural homology to known Pex3 proteins by blastp (protein–protein BLAST) analysis (https://blast.ncbi.nlm.nih.gov/Blast.cgi) (Altschul et al, 1990) was unsuccessful. We, therefore, used the protein comparative tool for remote protein homology detection and structure prediction, HHpred (Söding et al, 2005), to interrogate a *T. brucei gambiense* genome database with known Pex3 proteins. This HHpred interrogation identified one protein of unknown function, Accession no. XP_011780297.1 encoded by the Tbg972.11.11520 gene, with greater than 95% probability of being structurally homologous to the query human Pex3 protein (Fig 1A). The *T. brucei gambiense* protein identified related proteins from other kinetoplastids in the Eukaryotic Pathogen Database (Table S2). For expedience, these proteins were designated putatively as Pex3 proteins.

We performed primary structural alignments between the putative Pex3 proteins of *T. brucei brucei* strain TREU927 (Accession no. XP_829090.1, hereafter TbPex3, encoded by the Tb927.11.10260 gene [see Table S2]), *Leishmania major* strain Friedlin (XP_001686984.1, hereafter LmPex3, encoded by the LmjF.36.4010 gene [see Table S2]), and known Pex3 sequences from human (*Homo sapiens*, Hs), the plant *Arabidopsis thaliana* (At), the yeast *Saccharomyces cerevisiae* (Sc), the amoeba *Planoprotostelium fungivorum* (Pf), and the stramenopile *Nannochloropsis gaditana* (Ng). Overall sequence identity between the putative trypanosomatid Pex3 proteins and known Pex3 proteins was low, between 9.46 and 14.32% (Fig 1B and Table S1), thus providing an explanation for our inability to identify trypanosomal Pex3 by conventional BLAST search. However, all Pex3 protein sequences revealed a region of strong amino acid conservation around an aromatic residue corresponding to phenylalanine 102 (F102) in TbPex3, tryptophan 104 (W104) in HsPex3, and tryptophan (W) in all other Pex3 proteins (Fig 1B; residues in red and boxed).

### TbPex3 binds to the trypanosomal peroxisomal membrane protein receptor, TbPex19, via conserved amino acids

Pex3 proteins are integral to the peroxisomal membrane and have a cytosolic domain that docks the peroxisomal membrane protein receptor, Pex19 (Sacksteder et al, 2000; Fang et al, 2004). The nature of the Pex3–Pex19 interaction has been studied extensively in yeast and mammalian cells, and the crystal structure of HsPex3 in association with an HsPex19 peptide that corresponds to a region shown to interact with Pex3 has been solved (Sato et al, 2010; Schmidt et al, 2010). The HsPex19-binding region on the surface of HsPex3 forms a large hydrophobic groove into which the side chain of W104 protrudes. This aromatic ring intercalates between two "teeth" in the HsPex19 helix and is critical for binding of HsPex19 to HsPex3. Mutation of this tryptophan residue to alanine (W104A) abrogated the interaction

between HsPex3 and HsPex19 (Sato et al, 2008); however, of note, when W104 was replaced by an aromatic phenylalanine residue (W104F), the peroxisome-biogenic capacity of Pex3 was retained (Sato et al, 2008). The presence of an aromatic residue at position 104 is, therefore, necessary for the binding of HsPex3 to HsPex19 and, consequently, for peroxisome formation. Other conserved residues on the surface of HsPex3 in the vicinity of W104 contribute to its interaction with HsPex19 (Schmidt et al, 2010).

To ascertain whether the newly identified TbPex3 acts like a bona fide Pex3 protein, we tested its ability to interact with TbPex19 (Banerjee et al, 2005) and studied the effect of mutating amino acid residues that are conserved in known Pex3 proteins and are predicted to mediate the interaction between Pex3 and Pex19. TbPex3 did not interact with itself in a yeast two-hybrid assay (Fig 2A; top panels) but interacted with TbPex19 (Fig 2A; middle panels) at a strength comparable with that of the interaction between ScPex3 and ScPex19 (Fig 2A; bottom panels). TbPex3 and ScPex3 exhibited similar behavior when amino acids in the immediate vicinity of the predicted binding pocket for Pex19 were mutated. Significantly, TbPex3-F102A showed no interaction with TbPex19 (Fig 2A; middle panels), and mutation of the corresponding residue of ScPex3 (W128A) also abolished its interaction with ScPex19 (Fig 2A; bottom panels). Likewise, TbPex3-L105A and ScPex3-L131A did not interact with their respective Pex19-binding partners (Fig 2A). The corresponding leucine residue in HsPex3 (L107) had previously been identified as contributing to the binding of HsPex19 (Schmidt et al, 2010). On the other hand, TbPex3-K98A and ScPex3-K124A were unimpaired in their capacity to interact with their Pex19 partners (Fig 2A). TbPex3 apparently exhibited less stringency in its binding requirement for Pex19 outside of the immediate Pex19 binding region, as mutations of L89 and tyrosine 118 (Y118) to alanine did not affect the ability of TbPex3 to interact with TbPex19, whereas mutations of the corresponding residues in ScPex3 (L110A and Y144A) abrogated the interaction of ScPex3 with ScPex19 (Fig 2A).

Pull-down assays demonstrated that TbPex3 binds to TbPex19 and that mutations F102A and L105A of TbPex3 abolished this interaction (Fig 2B), in agreement with the yeast two-hybrid data (Fig 2A). We conclude that the newly identified TbPex3 acts like a canonical Pex3 protein in its interactions with the Pex19 protein.

### TbPex3 is a glycosomal protein

TbPex3, which is encoded by the gene Tb927.11.10260 of *T. brucei brucei*, had previously been shown to localize to glycosomes in a global fluorescent tagging project of all *T. brucei* proteins (Dean et al, 2017) and was ranked with high confidence as being glycosomal in two proteomic analyses of highly enriched glycosomes from *T. brucei* (Colasante et al, 2013; Güther et al, 2014). In agreement with these studies, indirect immunofluorescence analysis demonstrated that TbPex3 tagged at its C terminus with the hemagglutinin (HA) epitope (Pex3-HA) colocalized preferentially with the glycosomal matrix enzyme aldolase in cells of both the procyclic form (PCF) and the bloodstream form (BSF) of

## A

**XP_011780297.1** hypothetical protein, conserved [*Trypanosoma brucei gambiense* DAL972]

Probability: 95.12

```
Q ss_pred        HHHHHHHHHHCCHHHHHHHHc--cCcchHHHHHHHhhHHHHHHHHHHHHHHHHHHHHH------HHHHHHHhccc
Q HsPex3      74 PTLREALMQQLNSESLTALLK--NRPSNKLEIWEDLKIISFTRSTVAVYSTCMLVVL------LRVQLNIIGGY 139
Q Consensus   74 p~l~~~i~~~~dve~l~~~L~--~~~~~~K~eLWeeLKi~sftR~~~~iY~~~lL~l~------rvQlnILgr~ 139
                 +.-+..|.+.+.+.++.++||   .+++.|+..|.++=.++++-++++.|+.||-+.+      +++-+-+|+|+
T Consensus   70 ~ltK~L~~k~l~i~~lkeqLR~at~reaKl~~~~elF~ltLa~IlsaayvHSLtItLhviKn~~~VLVFvL~kR 146
T XP_011780297.1 70 PLTKSLMIDALAIGAIKERLRRATTPGDKVAIFSELFNTTLAGILSAAYIHSLTITLHVMRHTMSLLVFVLGKK 146
T ss_pred        HHHHHHHHHhchHHHHHHHHHhcCCchHHHHHHHHhhhHHHHHHHHHHHHHHHHHHHHHHHhHHHHHHHHHchh

Q ss_pred        hhhhhcccCCCC----------------------------------------CcccCCHHHHHHHHHHHHHHHhc
Q HsPex3          IYLDNAAVGKNG----------------------------------------TTILAPPDVQQQYLSSIQHLLGD 174
Q Consensus       ~y~~~~~~~~~~----------------------------------------~~~~~~~~~~~fLs~~~~~l~~ 174
                 --.........+                                        ......+..+.|  +.+-+++.
T Consensus       Ea~~~~r~s~~~~~S~lRsWW~~Gtq~~MMes~l~~m~qq~eaSPF~~~~~ee~~~~~dv~~~F--sV~alL~~ 215
T XP_011780297.1  EASSGTRRADTSLLSKFRSWWHQGTQNFMLQTLLENMTQQMEASPFSLMMDDDNMKTEDVKQGF--CVDALLRL 215
T ss_pred        hhcccccccCCCChhHHHHHHHHhHHHHHHHHHHHHHHHHHHHHHHHHHhChhhhhcccccccccchhhcCe--eHHHHHHh

Q ss_pred        cHHHHHHHHHHHHHHHHHhcC-----CCcccccCCHHHHHHHHHHHHHHHHHc
Q HsPex3          GLTELITVIKQAVQKVLGS-----VSLKHSLSLLDLEQKLKEIRNLVEQ 218 (373)
Q Consensus       G~~~L~~~V~~~V~~~l~~-------~~lk~~ls~~~l~~li~~Ir~~ie~ 218 (373)
                 -+++.+++....+|...++.      .++...++..|+.++++++...+|.
T Consensus       AvPRiv~aA~~vV~tal~~Rp~h~FsvtG~Vt~~dlr~Ll~ela~~fes 264 (474)
T XP_011780297.1  AVPRVVDTALHVVNTTLEGRDSQLFNLTGSVTEGEMRDLLQTLCAKFVS 264 (474)
T ss_pred        HHHHHHHHHHHHHHHHHccCchhhcccccccHHHHHHHHHHHHHHHHHhH
```

## B

```
Hs  -------------------------------MLRSVWNFLKRHKKKCIFLGTVLG--GVYILGKYGQKKI-REI--------QEREAAEYIAQARRQYHFESNQRTCNMTVLS-MLP  74
At  -------------------------------MDFVRGFW---RKHRRKVLVTAGCLG--SGYLLYKLYNSHT-RRLADLERELAHERRNDEII-KTQMKAHFESIQMIVDSTTLPHAMQ  81
Sc  -------------------------------MAPNQRSRSLLQRHRGKVLISLTGIA--ALFTTGSVVVFFV-KRW--LYKQ--QLRITEQHFIKEQIKRRFEQTQEDSLYTIYE-LLP  80
Pf  MLEDLAGLLGDSTQWTVANLYIVGQKLQQTGSSVGAW--VHKHRRKLIVVGILTG--SGFLAIRFIKYKI-NQF----KI--SFEKQQKEFDKMRMKNYFDSAQKSCDASVIN----  102
Ng  -------------------------------MGRLQQVYHFLKRNRRAIGLIAGASA--AGFAAWRI-KRVL-DDY---KRI--IREHDLARLDQHRLQMHMLRSRGECVPALLN-FMQ  78
Lm  ---------------M--DSLPNQ--------------LMSRITEISLSDAQIQSYGRWLAAAMGAYYVVNAW---------QKDTPADVQAAITGHMVQGYDRSTITTRLE--QR  74
Tb  ---------------MCDEFFGDL--------------LEEELENLGKAAFVVGVSGFLLKQAIGA----RSF---------LRGDP--VQNSIVAHLPQGYDRSIIAARVS--QP  70

Hs  TLREALMQ-QLNSESLTALLKN-----RPSN-------------KLEIWEDLKIISFTRSTVAVYSTCMLVVLLRVQLNIIGGYIYLDNA-AVGKNG---TTILAPP----------  158
At  FLSIRISE-EIDVSHVMDRINQGKGMLSPPE------------KLQLWDLIKILSFTRMVLSLWSVTMLSLYIRVQVNILGRHLYVDTARALGSSHLLCTLLISSPTSYFSGKLTGC  185
Sc  VWRMVLNENDLNLDSIVTQLKDQKNQLTRAKSSESRESSPLKSKAELWNELELKSLIKLVTVTYTVSSLILLTRLQLNILTRNEYLDSAIKLTMQQENCNKLQNRFYNWVTSWWSDP  197
Pf  FIEHALRN-RLNTAVEIPSAQAVRQIESKEE------------KLAQWEKIKVGSFTKALLTLYTVTLLTLFVRVEVNILGRYVYMNTAISVTDEG---------------------  185
Ng  TLRKRVYE-IVDVTSPVKALKAGRGGLSKQE------------FQALWHQVKVSGFSRFFLAYMGFNLLNVMLRVQVHILGRYAFEASRQEMLQAAQRDQEEGQGPRGEFCS-LGDA  181
Lm  HTRQLLQQ-IIGLNELRQQIRSET---DREA------------KLRGWQRVFRLTLISIAATTYTHSLTISLLSIK-NMVRVLVFLLSRRE-SAAL---SKSRSGPISTLRAWWTGG  170
Tb  LTKSLMID-ALAIGAIKERLRRAT---TPGD------------KVAIESLFNTTLAGILSAAYIHSLTITLHVMR-HTMSLLVFVLGKQEASSGT---RRANTSLLSKFRSWWHQG  167

Hs  ------------------------------------DVQ-----------------------------------QQYLSSI-QHLLGDGLTELITVIKQAVQKVLGS-  193
At  SVKNF-------------------SRLPFKATAACPEEVDLIDRDDE-------------------------------QKFLSSA-DFLVTNAMPSLISDMQGSAEEVLKG--  245
Sc  EDKAD-------------------DAMVMAAKKSKKEGQEVY-INE-------------------------------QAFLSLS-WWILNKGWLSYNEIITNQIBIEFDG-  255
Pf  ----D------------------DQMPVEEPIPD--------AIT-------------------------------KKYLAHT-EYLIEEGLPELAEFITSQVEEEMTA-  232
Ng  GYAAT------------------GSSSFGS------------DDR-------------------------------CRLLSLVYEHFLGEGLRRLKEAVEVAVREELGA-  229
Lm  RRAGLTGIMMESMMAKMAEQVSQQASPFVNAELEEEVVTE----EVRQLPVPPPPLDGRTAFAQAQQQHEMLEAAHGESQSQVEQAFSVRA---VLEVAVPRIVACAAAAVDSAIAAR  281
Tb  TQ----NFMLQTLLENMTQQM--EASPFSLMMDDDNMKTE----DVK-------------------------------QGFCVDA---LLRLAVPRVVDTALHVVNTTLEGR  235

Hs  ----VSIKHSLSLL--DLEQKLKEIRN-LVEQHKSSSWINKDSGSKPLLCHYMMPDEETP---------LAVQACGLSPRDITTI-KLLNETRDMLESPDFSTVLNTC-----LNRGF  288
At  ----KQLKDVITTR--VLQETVMQIVDVFMSTGSPHHWVD----------YLMMPQDTK--------LSRTTSDSSDEAVSKFHQLMVETREVLISTEFTNIVEIS-----L-KCF  331
Sc  ----IHPRDTLTLE--EFSSRLTN-----IFRNTNSQIFQQNNNNLT----SILLPKDSSGQEFLLSQTLDADALTSFHSNTLVFNQLVNELTQCIESTATSIVLESL-----INESF  353
Pf  ----WPVTGKYNTE--HLSALLANLRN--------RIDFLPGGMQRRPLYPYLLPVEEDL-----------IEESLPEQLQ-------CLLNETRDVTESEKFGNILGVM-----LDQAC  315
Ng  ----WSVHAKIHVDYAELHDALMRIRR-RLEGPRGVLLAE-----SPLLQYVLDAASQA--------------EDLGPPGSRLHEMVNETWDAFESPSFKLAVEDC-----LDVTF  316
Lm  PAHLFSVSGIVRAV--DLCSLLRDI---ALGMERRACVSE----------WVHRSHPSI-------VPMPRARSGASISPLASLPKQLETSKGRLPPDEGEELLDEPDQ----SATSS  373
Tb  DSQLFNLTGSVTEG--EMRDLLQTLI---CAKFVSRATLSD----------WLTPPSEEL----------KDSDGGVLNTSPPPQGMGDGESGGESPGGRDKAMSNGDSLMSNDEQS  326

Hs  SRLLDNM------------------------------------------AEFFRPTEQDLQHG--NSMNSLSSVSLPLAKIIPIVNGQIHSVCS-E-TPSHFVQDLLTMEQVKDF  357
At  TDVL------------------------------------------VEEMETQ--TEAGGLAT-GKPLAKVLPQIEKTMNVITA-EPSKNRFLQIIRDLPEVKLF  390
Sc  HFIMNKV------------------------------------------GIKTIAKKKPGQED--QQQYQMAVFAMSMKDCCQEMLQTTAGSSH-SGSVNEYLATLDSVQPLDDL  423
Pf  SSLLNGI------------------------------------------SAAYQ-----TSN--QVNNGL----MAMPHIIPILRDQASKLID-EKVPENVLWPLFKNQQVEEY  375
Ng  RVFELEDLFRSLYASPPSASRSARPGADEKERPLGEGIGEGPVAERSTEEEGSEEARDSDADRAGA--EESKPLDP---PLAKLIPHMKNAATKVFNGDPQNNEYIRITAQNRSVNLL  428
Lm  EDIIE------------FPHGSTTLLRPD----------------------------GRPVRRRAGDRSED--KILDGLLNRRLTAEEEALQEETIRRDHLL-KRQMAGFFTELSHSASFGEL  453
Tb  EDRLG--------LGGGGG-----G------------------------------GMPLQGVFPPLSANFFPMANGAAH---AEEMHTAHRNRLIHEKLR-RERAAGFFREIVHSVSLSEL  400

Hs  AANVYEAFSTPQQLEK---------------------------------------------------  373
At  FTLLYANMPQ-------------------------------------------------------  400
Sc  SASVYSNFGVSSFSFKP--------------------------------------------------  441
Pf  SYYIFTSSYDQE------------------------------------------------------  387
Ng  CSSFFTIETSFD------------------------------------------------------  440
Lm  CITYAAELLAAQ-FKE-ACDIKRVKTYDSATDTAKMAMVIAALDSCRLAALEEECEVKSYMRLFCEETIRSTCAR-  526
Tb  CIAYTEELLGAA--VA-TTDFSSLRPEGDSTVAVRVPQMLPLLEKQRLDMFDCDFQVVQPYVRLLCEETIRVTCRDL  474
```

**Figure 1. Bioinformatics-based identification and features of putative trypanosomatid Pex3 proteins.**
**(A)** The secondary structure-based homology program HHpred was used to search a database of *T. brucei gambiense* with the known HsPex3 sequence. The uncharacterized target (T) protein XP_011780297.1 encoded by the Tbg972.11.11520 gene was identified with greater than 95% probability as having a similar predicted secondary structure (ss pred) as the query (Q), HsPex3. HHpred output includes a prediction of helices (H, h) and coils (C, c) (capitals represent a more reliable

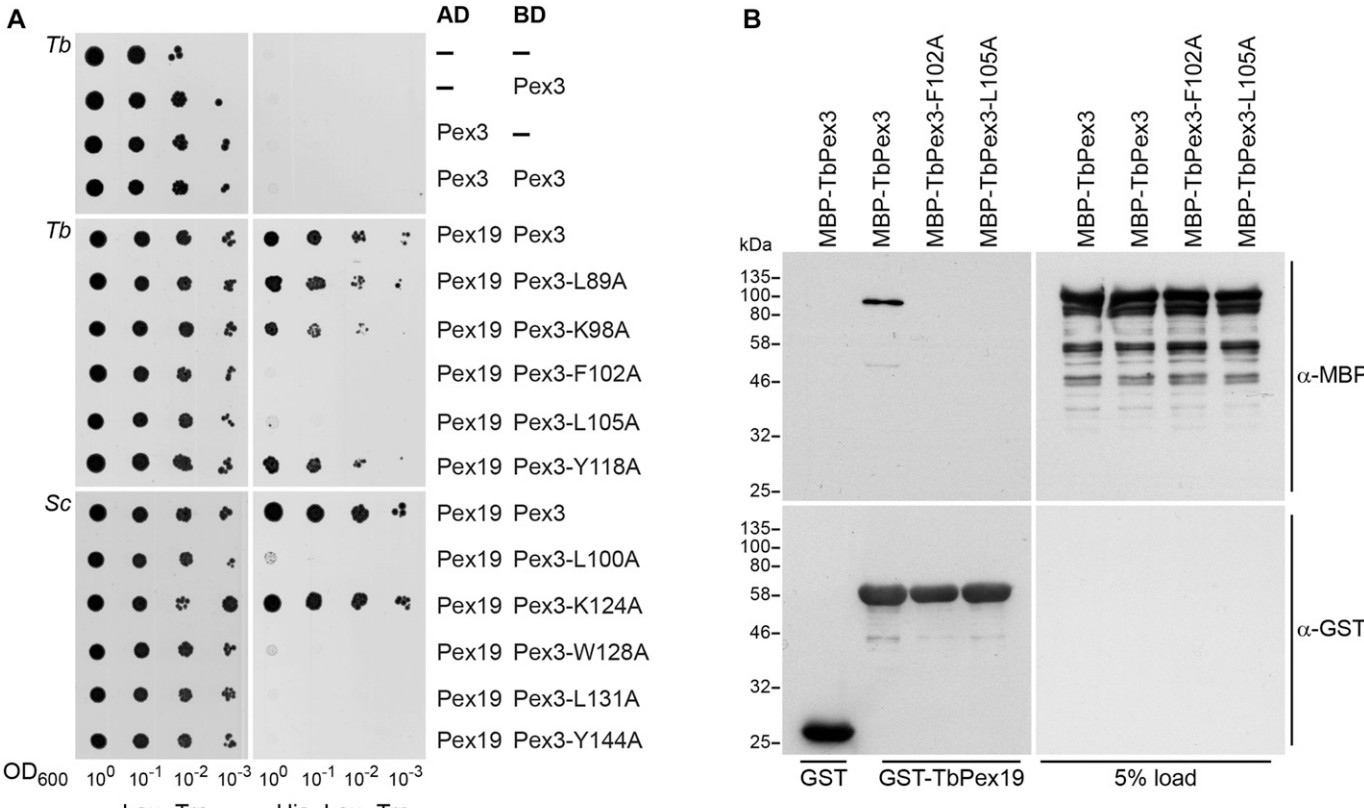

**Figure 2. TbPex3 displays an orthodox mode of binding to TbPex19.**
**(A)** Yeast two-hybrid analysis of protein interactions between TbPex3 and TbPex19, and between ScPex3 and ScPex19. *S. cerevisiae* HF7c cells expressing Gal4-AD and Gal4-BD protein fusions to wild-type and point mutants of Pex3 and wild-type Pex19 were grown in liquid synthetic dropout medium and adjusted to an OD$_{600}$ of 1.0. A 10-fold serial dilution series was spotted onto –Leu –Trp (left) and –His –Leu –Trp (right) plates. Growth on –Leu –Trp medium requires cells to have both AD and BD plasmids and is indicative of cell number. Growth on –His –Leu –Trp medium occurs only when there is a protein–protein interaction. **(B)** TbPex3 binds TbPex19. GST alone or GST-TbPex19 fusions immobilized on glutathione-sepharose beads were incubated with extracts of *E. coli* expressing either MBP-TbPex3, MBP-TbPex3-F102A, or MBP-TbPex3-L105A. Bound protein was detected by immunoblotting with anti-MBP antibody (upper panel, left). Total GST fusion proteins were visualized by immunoblotting with anti-GST antibody (lower panel, left). Panels at right show 5% of load. Numbers at left denote migration of molecular mass markers.

*T. brucei* (Fig 3A). Pex3-HA also localized to reticular structures devoid of glycosomal aldolase in cells (Fig 3A). In addition, subcellular fractionation and immunoblot analysis showed co-fractionation of Pex3-HA with the glycosomal enzymes aldolase and GAPDH and to a lesser extent with the ER marker BiP, but not with the mitochondrial enzyme lipoamide dehydrogenase (Fig 3B). These observations are consistent with results from previous studies showing that Pex3 is a peroxisomal protein that passages through the ER during the de novo formation of peroxisomes in both yeast and mammalian systems (for a review, see Farré et al (2018)).

**Depletion of TbPex3 leads to defective glycosomes and death of both trypanosomal PCF and BSF cells**

To examine the role of TbPex3 in the maintenance of the glycosomal compartment in vivo, we used RNAi (Wickstead et al, 2002) to deplete the transcript for TbPex3 from cells of *T. brucei*. A stem-loop construct of the *TbPex3* gene was transfected into both PCF and BSF cells, and analysis was performed on uninduced (no tetracycline, –Tet) and RNAi-induced (with tetracycline, +Tet) PCF and BSF cell cultures. Pex3-specific mRNA was no longer detectable in RNAi-induced PCF and BSF cells on day 3 of culture (Fig 4A).

prediction), of conserved consensus residues (capitals represent a more reliable prediction) calculated from the query (Q) or Target (T), and a strength of alignment of amino acids according to their biophysical properties ("|" very good, "+" good, "·" neutral, "−" bad, and "=" very bad) (Söding et al, 2005). **(B)** Alignment of the putative trypanosomatid Pex3 sequences from *Leishmania major* (Lm; Accession no. XP_001686984.1) and *T. brucei* (Tb; Accession no. XP_829090.1) with known Pex3 sequences from human (Hs; Accession no. NP_003621.1), the plant *A. thaliana* (At; Accession no. NP_001154410.1), the yeast *S. cerevisiae* (Sc; Accession no. NP_010616.3), the amoeba *P. fungivorum* (Pf; Accession no. PRP89031.1), and the stramenopile *N. gaditana* (Ng; Accession no. EWM25801.1). Sequences were aligned using Kalign (https://www.ebi.ac.uk/Tools/msa/kalign/). An amino acid that is identical to its corresponding amino acid in TbPex3 is highlighted in blue. An amino acid that is similar to its corresponding amino acid in TbPex3 is highlighted in yellow. Groupings of similar amino acids are as follows: (G, A, S), (A, V), (V, I, L, M), (I, L, M, F, Y, W), (K, R, H), (D, E, Q, N), and (S, T, Q, N). Dashes represent gaps. The conserved aromatic residues W/F of Pex3 required for interaction with Pex19 are shown in red and boxed. Other conserved residues that were mutated to evaluate the effect on the interaction between TbPex3 and TbPex19 in comparison with the interaction between ScPex3 and ScPex19 are boxed. See also Fig 2.

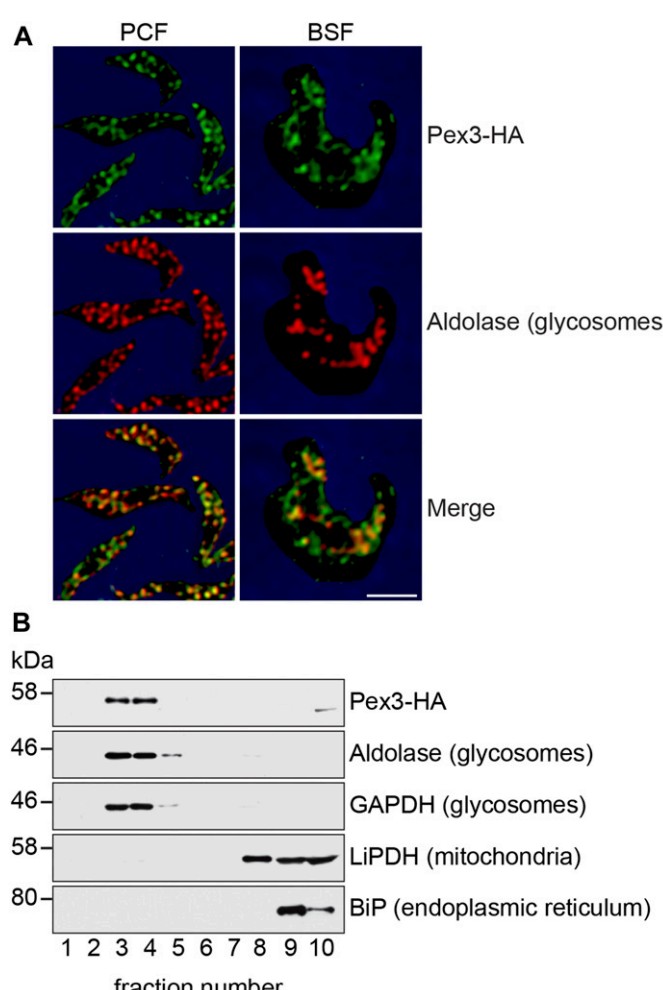

**Figure 3. TbPex3 is a glycosomal protein.**
**(A)** TbPex3 is a glycosomal protein. PCF and BSF cells expressing TbPex3 C-terminally tagged with the HA epitope were fixed with formaldehyde and processed for immunofluorescence microscopy with mouse anti-HA antibodies and Alexa Fluor 488 rabbit antimouse IgG (top panels, green) and with rabbit anti-aldolase antibodies and Alexa Fluor 568 goat antirabbit IgG (middle panels, red). Merged images are presented in bottom panels. Bar, 2 μm. **(B)** TbPex3 cofractionates with glycosomal enzymes. Postnuclear lysates of PCF cells expressing Pex3-HA were fractionated on a discontinuous sucrose gradient. Equivolume fractions collected from the bottom of the gradient were analyzed by immunoblotting with antibodies to HA, aldolase, GAPDH, lipoamide dehydrogenase, and BiP.

RNAi reduction of *TbPex3* transcript in both PCF and BSF cells resulted in a largely diffuse immunofluorescence signal for glycosomal aldolase (Fig 4B, +Tet), indicative of its mislocalization to the cytosol, which contrasted with the generally punctate appearance of aldolase in uninduced PCF and BSF cells (Fig 4B, –Tet). Increasing numbers of aberrant glycosome structures, glycosome fragmentation, and mislocalization of glycosomal aldolase to the cytosol observed by immunofluorescence microscopy paralleled the progressive reduction of the *TbPex3* transcript over time (Fig S1), consistent with a role for TbPex3 in glycosome biogenesis. The compartmentalization of glycosomal matrix proteins was also assessed biochemically by differential digitonin permeabilization of uninduced and induced cultures of PCF cells (Fig 4C). Digitonin

selectively permeabilizes the plasma membrane, therefore preferentially liberating cytosolic proteins rather than proteins enclosed by an organellar membrane to a supernatant fraction after centrifugation. The cytosolic protein tubulin was released to the supernatant at the lowest concentrations of digitonin in both uninduced (–Tet) and induced (+Tet) PCF cells. In contrast, the glycosomal matrix enzymes aldolase and GAPDH were released at lower concentrations of digitonin in RNAi-induced PCF cells (+Tet, 0.02 mg digitonin per mg protein) versus uninduced cells (–Tet, 0.16 mg digitonin per mg protein).

Glycosomes were also visualized by EM of uninduced and RNAi-induced PCF and BSF cells (Fig 4D). In electron micrographs, glycosomes appear as round, electron-dense organelles of essentially uniform size that are surrounded by a single unit membrane (Banerjee et al, 2005; de Souza, 2008). Quantitative analysis of EM images revealed ~13-fold and 7-fold reductions in the number of glycosomes per cell volume in RNAi-induced PCF cells and BSF cells, respectively, in comparison with their corresponding controls (Fig 4D and Table 1). On average, glycosomes were larger by 35 or 48% in RNAi-induced BSF cells and PCF cells, respectively, than in their corresponding uninduced cells (Table 1), which is indicative of a glycosome assembly defect and suggestive of a redirection of matrix proteins to existing glycosomes. Collectively, our data show that Pex3 is essential for the maintenance of the glycosomal compartment in *T. brucei*.

We investigated the effect of reducing TbPex3 abundance by RNAi on the viability of PCF and BSF cells (Fig 4E). Uninduced (–Tet) and RNAi-induced (+Tet) PCF cells grew similarly for 8 d from the start of RNAi. Between days 8 and 9, the viability of RNAi-induced PCF cells fell precipitously, and no viable RNAi-induced PCF cells were seen 9 d after the start of RNAi. In contrast, uninduced PCF cells continued their growth unabated. RNAi-induced BSF cells grew essentially like uninduced BSF cells for 4 d, when again the viability of RNAi-induced BSF cells fell precipitously so that essentially all RNAi-induced cells were dead 5 d after the start of RNAi (Fig 4 E). Again, uninduced BSF cells continued their unabated growth. Survival of PCF and BSF cells for a number of days after the start of RNAi treatment could be due to sufficient glycosomal activity being maintained for a period of time despite reduced numbers of glycosomes, increased glycosome fragmentation, and mislocalization of glycosomal enzymes to the cytosol, until a tipping point in glycosome functionality is reached and a precipitous reduction in cell survival occurs. Our results are consistent with the findings of a global RNAi analysis in which *T. brucei* BSF cells knocked down for the *TbPex3* gene did not show a significant loss of fitness after 3 d of RNAi treatment but did show a significant loss of fitness after 6 d of treatment (Alsford et al, 2011).

In summary, we have identified the first trypanosomatid Pex3 through the use of the HHpred bioinformatics platform that looks for similarities in protein secondary structure rather than for similarities in protein primary structure. Despite its divergence in primary structure from characterized Pex3 proteins, TbPex3 nevertheless acts like a canonical Pex3 in that it binds Pex19 through conserved residues, which also mediate the interaction between Pex3 and Pex19 in other organisms. Like the necessity for Pex3 proteins in peroxisome biogenesis, TbPex3 is necessary for glycosome biogenesis. Reduction in TbPex3 amounts led to reduced numbers of enlarged glycosomes in both PCF cells and BSF cells

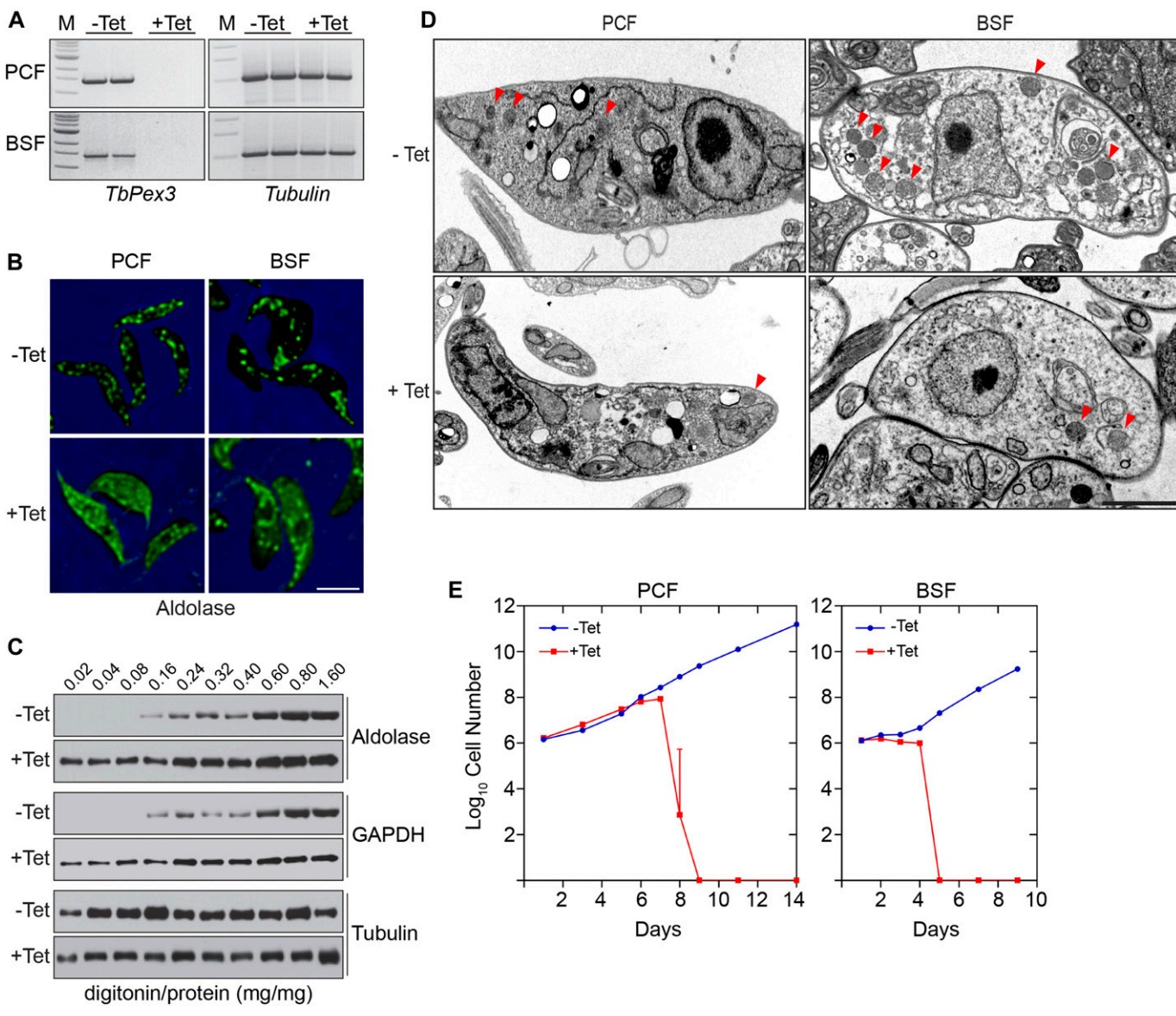

**Figure 4. TbPex3 is essential for glycosome biogenesis and trypanosome viability.**
**(A)** RNAi eliminates the *TbPex3* transcript. Semiquantitative reverse transcription polymerase chain reaction (RT-PCR) analysis of transcript levels for *TbPex3* and *Tubulin* (control) from uninduced (–Tet) and RNAi-induced (+Tet) PCF and BSF cells of *T. brucei* on day 3 of culture. **(B)** *TbPex3*-RNAi leads to mislocalization of the glycosomal matrix enzyme aldolase to the cytosol. Uninduced (–Tet) and RNAi-induced (+Tet) PCF and BSF cells of *T. brucei* on day 3 of treatment were processed for immunofluorescence microscopy with rabbit anti-aldolase antibodies and Alexa Fluor 488 goat antirabbit IgG. Bar, 2 μm. **(C)** *TbPex3*-RNAi–treated cells release glycosomal matrix enzymes at lower amounts of the detergent, digitonin, than do uninduced cells. Aliquots of untreated (–Tet) and RNAi-treated (+Tet) *T. brucei* PCF cells were incubated with increasing concentrations of digitonin and subjected to centrifugation. Supernatants were analyzed for the presence of aldolase, GAPDH, and tubulin. **(D)** Transmission EM of *TbPex3*-RNAi cells. Uninduced (–Tet) and RNAi-induced (+Tet) PCF and PSF cells of *T. brucei* were visualized on day 3 of treatment. Red arrowheads point to glycosomes. Bar, 5 μm. Quantification of EM images is presented in Table 1. **(E)** Pex3 is essential for cell viability of *T. brucei*. Growth curves of uninduced (–Tet) and *TbPex3*-RNAi–induced (+Tet) PCF cells and BSF cells. Error bars present SEM of triplicate readings.

and to the mislocalization of glycosomal matrix enzymes to the cytosol. Notably, reduction in the amounts of Pex3 led to the death of all PCF and BSF cells following *Pex3*-RNAi. This absolute lethality stands in contrast to the results of other studies in which cell growth recovered after an initial drop in cell numbers after RNAi reduction of a trypanosomal *Pex* gene transcript (Banerjee et al, 2005; Krazy & Michels, 2006; Kalel et al, 2015). We conclude that *TbPex3* is an essential gene in *T. brucei*.

# Materials and Methods

### Identification of a candidate TbPex3

Attempts at identifying a candidate trypanosomatid Pex3 by conventional blastp (protein–protein BLAST) analysis of known Pex3 proteins as queries were unsuccessful. HHpred is a rapid interactive server for protein homology detection and structure detection. We

**Table 1. Glycosome number and average area in uninduced cells and *TbPex3*-RNAi–induced cells.**

| Form (number of cells analyzed) | Glycosome number/cell volume ($\mu m^3$) | Glycosome average area ± SEM ($\mu m^2$) |
|---|---|---|
| PCF uninduced (n = 99) | 0.078 | 0.46 ± 0.15 |
| PCF induced (n = 99) | 0.006 | 0.68 ± 0.36 |
| BSF uninduced (n = 103) | 0.078 | 0.78 ± 0.26 |
| BSF induced (n = 103) | 0.011 | 1.05 ± 0.31 |

used HHpred (https://toolkit.tuebingen.mpg.de/#/tools/hhpred) to query the *T. brucei gambiense* genome with human and yeast *S. cerevisiae* Pex3 proteins. Both queries identified the *T. brucei gambiense* protein XP_011780297.1 of unknown function encoded by the Tbg972.11.11520 gene as a potential trypanosomatid Pex3 homologue with greater than 95% probability. Subsequent experimentation demonstrated the homologous protein XP_829090.1 of *T. brucei brucei* encoded by the Tb927.11.10260 gene to be the Pex3 protein of *T. brucei*.

### Trypanosome cell culture and transgenic lines

PCF cells of *T. brucei* Lister 427 29-13 (TetR T7RNAP), which co-express the tetracycline (Tet) repressor and T7 RNA polymerase, were maintained in SDM-79 medium (Invitrogen) containing 10% fetal bovine serum, 50 $\mu g$ hygromycin ml$^{-1}$ and 2.5 $\mu g$ G-418 ml$^{-1}$, at 25°C with 5% $CO_2$ in a water-saturated incubator. Transgenic lines were generated by the limiting dilution method (Krazy & Michels, 2006) and selection with 2.5 $\mu g$ phleomycin ml$^{-1}$. For RNAi studies, PCF cells were grown in glucose-free medium (Wickstead et al, 2002).

BSF cells of *T. brucei* Lister 427 VSG 221 (TetR T7RNAP) were maintained in HMI-9 medium containing 10% fetal bovine serum, 10% Serum Plus (Sigma-Aldrich), and 2.5 $\mu g$ G-418 ml$^{-1}$. The cultures were maintained at 37°C with 5% $CO_2$ in a water-saturated incubator. Transgenic lines were obtained by the limiting dilution method and selection using 2.5 $\mu g$ phleomycin ml$^{-1}$.

### Yeast two-hybrid analysis

PCR products encoding full-length TbPex3 and mutants TbPex3-L89A, TbPex3-K98A, TbPex3-F102A, TbPex3-L105A, and TbPex3-Y118A and PCR products encoding amino acids 51–441 of wild-type *S. cerevisiae* (Sc) Pex3 and mutants ScPex3-L100A, ScPex3-K124A, ScPex3-W128A, ScPex3-L131A, and ScPex3-Y144A were cloned in-frame and downstream of the DNA-binding domain (BD) of the *GAL4* transcriptional activator in pGBT9 (Clontech). Full-length TbPex3, TbPex19, and ScPex19 were cloned in-frame and downstream of the activation domain (AD) of the *GAL4* transcriptional activator in pGAD424 (Clontech). Plasmids were transformed into *S. cerevisiae* strain HF7c, and transformed cells were grown on synthetic dropout medium agar lacking leucine and tryptophan (−Leu −Trp) to determine total cell growth and on synthetic dropout medium agar lacking histidine, leucine, and tryptophan (−His −Leu −Trp) to determine the growth of cells exhibiting protein–protein interaction between the AD-fusion and BD-fusion constructs.

### Assay for protein binding

Binding between TbPex3 and TbPex19 was examined essentially as described (Knoblach et al, 2013). GST fusion to TbPex19 was constructed in pGEX4T-1 (GE Healthcare). MBP fusions to TbPex3 and to mutants TbPex3-F102A and TbPex3-L105A were constructed in pMAL-c2 (New England Biolabs). Recombinant proteins were expressed in the *Escherichia coli* strain BL21 (Invitrogen). GST alone or GST-TbPex19 was immobilized on glutathione-sepharose beads and incubated with *E. coli* lysates containing MBP-TbPex3 or MBP-TbPex3-F102A or MBP-TbPex3-L105A in binding buffer (20 mM Tris–HCl, pH 7.5, 100 mM KCl, 5 mM $MgCl_2$, and 0.5% (vol/vol) Triton X-100). Unbound proteins were removed by washing five times in binding buffer. Immobilized proteins were eluted in sample buffer (50 mM Tris–HCl, pH 6.8, 2% SDS, 5% [vol/vol] glycerol, 0.002% bromophenol blue, 100 mM 2-mercaptoethanol) and subjected to SDS–PAGE and immunoblotting.

### Cloning of the *TbPex3* gene

The *TbPex3* gene (Tb927.11.10260) was amplified from genomic DNA using primers 5′-CCCAAGCTTATGTGTGACGAGTTCTTTGGAG and 5′-CGACTAGTTAAATCGCGGCATGTAACTCTAA. To tag TbPex3 with three copies of the HA tag at its C terminus, genomic DNA was amplified using primers 5′-CCCAAGCTTATGTGTGACGAGTTCTTTGGAG and 5′-GCGGGATCCTTAGGCGGCCGGAGCGTAATCTGGAACGTCATATGGAT-AGGATCCTGCATAGTCCGGGACGTCATACGGATAGCCCGCATAGTCAGG-AACATCGTATGGGTAAACGGCCGCTAAATCGCGGCATGTAACTCTAA, and the resultant product cloned into vector pDEX577-C (Kelly et al, 2007) for expression.

### Immunofluorescence microscopy

PCF and BSF cells were harvested by centrifugation, washed in PBS, pH 7.4, and spread onto coverslips coated with poly-L-lysine. The cells were fixed in PBS containing 4% paraformaldehyde for 10 min and permeabilized/blocked using 50 mM Tris–HCl, pH 7.5, 0.25% Triton X-100, and 2% FBS for 30 min. The cells were washed with PBS, incubated with primary antibody in PBS containing 2% FBS for 2 h, washed with PBS, and incubated with fluorescent secondary antibody (Alexa Fluor 488 rabbit antimouse IgG, Alexa Fluor 488 goat antirabbit IgG, or Alexa Fluor 568 goat antirabbit IgG) for 1 h. The cells were washed in PBS and mounted in 50% glycerol/*n*-propyl gallate. Images were acquired with an LSM710 confocal fluorescence microscope (Carl Zeiss) equipped with an oil immersion objective and ZEN 2009 acquisition software (Carl Zeiss).

Acquired fluorescence images were deconvolved using algorithms provided by Huygens Professional Software (Scientific Volume Imaging BV). 3D data sets were processed to remove noise and re-assign blur through an iterative Classic Maximum Likelihood Estimation algorithm and an experimentally derived point spread

function. The transmission image was treated differently. A Gaussian filter and blue color were applied to the transmission image using Imaris software (v.7.7.2) (Bitplane). The level of the transmission image was modified, and the image was processed until only the circumference of the cell was visible. Internal structures were removed in Adobe Photoshop to prevent interference by internal structures captured in the transmission images. Imaris software was then used to display the deconvolved data set with the processed transmission image and to prepare the image files before final figure assembly in Adobe Photoshop and Adobe Illustrator.

### Subcellular fractionation and immunoblot analysis

PCF cells were pelleted by centrifugation and washed once with PBS, pH 7.4, and once with homogenization buffer (25 mM Tris–HCl, pH 8.0, 250 mM sucrose, 1 mM EDTA, 1 mM DTT, and 1× complete protease inhibitors [Roche]). Washed cells were disrupted in homogenization buffer containing silicon carbide (400 mesh; Sigma-Aldrich) and subjected to centrifugation at 1,000$g$ for 10 min at 4°C to remove cell debris and unbroken cells. The postnuclear supernatant was loaded onto a 30-ml discontinuous sucrose gradient consisting of 25, 50, and 60% sucrose steps and a 70% sucrose cushion (all sucrose solutions, wt/vol) and subjected to centrifugation at 216,000$g$ for 90 min in a VTi50 rotor (Beckman) at 4°C. Equivolume fractions were collected, and proteins were precipitated by addition of trichloroacetic acid.

Immunoblot analysis was performed using standard protocols. Immunoreactive proteins were visualized with the SuperSignal West Femto Maximum Sensitivity Substrate chemiluminescence detection system (Thermo Fisher Scientific).

### RNAi and semiquantitative reverse transcription polymerase chain reaction (RT-PCR)

An RNAi stem-loop construct for the *TbPex3* gene was made by PCR amplification using primers 5′-GCGGGATCCGACCGGAGCATCATTGCCG and 5′-GGAATTCTGTAGGCACN$_{50}$GAGTGTCTGCAACATGAAATTC. The PCR product, which has a randomized 3′ end containing an EcoRI site, was digested with EcoRI and ligated to form a stem-loop region. The ligated product was purified, digested with BamHI, and inserted into vector p2T7-177 (Wickstead et al, 2002) at the BamHI site.

RNAi was induced in PCF cells by addition of tetracycline to 2 $\mu$g·ml$^{-1}$ final concentration in glucose-free medium essentially as described (Wickstead et al, 2002). After 2 d, glucose was added to 10 mM, and culturing continued until day 14. Parasites were diluted every 24 h, and fresh tetracycline and glucose were added to induced cultures.

RNAi was induced in BSF cells by addition of tetracycline to 2 $\mu$g·ml$^{-1}$ final concentration in HMI-9 medium containing 10% FBS, 10% serum plus, 2.5 $\mu$g G-418 ml$^{-1}$, and 2.5 $\mu$g phleomycin ml$^{-1}$ essentially as described (Wickstead et al, 2002). The growth of uninduced and RNAi-induced cultures was monitored for 9 d.

Total RNA was isolated from uninduced and induced PCF cells or BSF cells using Trizol reagent (Invitrogen) and then treated with DNase I. cDNA was made using reverse transcriptase and amplified by PCR using primers 5′-GGAATTCATGTGTGACGAGTTCTTTGGAG and

5′-GCGGGATCCTTATAAATCGCGGCATGTAACTC (Fig 4A) or primers 5′-GGAAAAGGGCCCGCAAAGCGAAGATCGGTTGGGGGCTGCTTCTCTA and 5′-GCAACGGTAG (Figs S1 and 1A) for the *TbPex3* gene or primers 5′-CACCTCGAGATGCGTGAGGCTATCTGCATC and 5′-CACAAGCTTTGGA-TACACCGTGTAGCCGAG for the gene for *α-Tubulin* (Tb927.1.2380).

## Supplementary Information

## Acknowledgements

The authors thank X Sun for help with confocal fluorescence microscopy, N Tahbaz for help with EM, K Tedrick for help with yeast two-hybrid analysis and pull-down assays, and A Simmonds for help with RT-PCR. This work was supported by Foundation Grant FDN-143289 from the Canadian Institutes of Health Research to RA Rachubinski.

### Author Contributions

H Banerjee: conceptualization, data curation, formal analysis, validation, investigation, methodology, and writing—original draft, review, and editing.
B Knoblach: conceptualization, data curation, formal analysis, validation, investigation, methodology, and writing—original draft, review, and editing.
RA Rachubinski: conceptualization, resources, formal analysis, funding acquisition, validation, investigation, methodology, project administration, and writing—original draft, review, and editing.

### Conflict of Interest Statement

The authors declare that they have no conflict of interest.

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
