## [Reviewer comments · Life Science Alliance]

Life Science Alliance

The early acting glycosome biogenic protein Pex3 is essential for trypanosome viability

Hiren Banerjee, Barbara Knoblach, and Richard Rachubinski

DOI: <https://doi.org/10.26508/lsa.201900421>

Corresponding author(s): *Richard Rachubinski, University of Alberta*

Review Timeline:

Submission Date:	2019-05-09
Editorial Decision:	2019-05-24
Revision Received:	2019-07-12
Editorial Decision:	2019-07-16
Revision Received:	2019-07-16
Accepted:	2019-07-17

Scientific Editor: Andrea Leibfried

Transaction Report:

May 24, 2019

Re: Life Science Alliance manuscript #LSA-2019-00421-T

Dr. Richard A. Rachubinski
University of Alberta
Cell Biology
Medical Sciences Bldg. 5-14
Medical Sciences Building 5-14
Edmonton, Alberta T6G 2H7
Canada

Dear Dr. Rachubinski,

Thank you for submitting your manuscript entitled "The early acting glycosome biogenic protein Pex3 is essential for trypanosome viability" to Life Science Alliance. The manuscript was assessed by expert reviewers, whose comments are appended to this letter.

As you will see, the reviewers appreciate your results and provide constructive input on how to further strengthen your work. We would thus like to invite you to submit a revised version of your manuscript, addressing the individual reviewer points raised. Importantly, the knockdown kinetics should get analyzed (rev#1 and rev#3) and the text should get re-written to remove the claims on Pex3 being a good drug target (rev#3). A few other points require your attention, but all seem straightforward to address.

Thank you for this interesting contribution to Life Science Alliance. We are looking forward to receiving your revised manuscript.

Sincerely,

B. MANUSCRIPT ORGANIZATION AND FORMATTING:

Reviewer #1 (Comments to the Authors (Required)):

Banerjee et al. have identified Pex3 in trypanosome species. Pex3 is known to orchestrate early

steps in peroxisome biogenesis, but the trypanosome ortholog has never been identified. The authors identified the candidate using HHPred. They showed that it localizes to glycosomes (the trypanosomal peroxisome analog) and that it binds to the peroxisomal chaperone Pex19, known to bind to Pex3 in other eukaryotes. They constructed strains in two species in which Pex3 was knocked down. In both strains glycosomal matrix proteins failed to localize, cells had few and aberrant glycosomes, and they failed to grow.

While there is no new insights here into peroxisomal assembly, the paper shows that this important protein and its function is conserved in trypanosoma. As cells require Pex3 for survival, Pex3 may be an important target for new drug development.

Data are straightforward and convincing. I have one significant (#1) and one minor (#2) comment:

(1) Some glycosomes are seen in the knockdown on day 3 of treatment, which is not expected in the absence of Pex3. Do you think that Pex3 is redundant or that some Pex3 protein still exists and functions? Can't the half-life of Pex3 in the knock-down be easily ascertained with the HA-tagged protein that the authors already generated? Are the number of glycosomes further reduced at day 6 or 7 before the cells die?

(2) Page 5 states that "...position 104 is therefore sufficient for the binding of HsPex3". Do the data only show that position 104 is necessary (but not necessarily sufficient)?

Reviewer #2 (Comments to the Authors (Required)):

Glycosomes are peroxisome-like organelles that are essential for the viability of *Trypanosoma brucei* (Tb). Although several proteins that contribute to the biogenesis and function of glycosomes were identified in the past, Pex3, a key protein that exists in peroxisomes in all eukaryotes, was not identified. In their manuscript, Banerjee et al. identified the Tb Pex3 protein by using the HHPred tool. They show that the putative Tb Pex3 is similar to well-studied Pex3 proteins in other organisms, that Tb Pex3 binds to Tb Pex19 (as do all other studied Pex3 proteins to date), and that reducing the amounts of Tb Pex3 drastically affects the formation of glycosomes. Moreover, they show that Tb Pex3 is essential for the viability of the parasite and hence suggest that Tb Pex3 is a potential drug target.

The data seem solid, the experiments well performed, the manuscript well written, the scientific story is well presented. Importantly, the identification of the Tb Pex3 is an important piece in the cell biology of this parasite and may contribute to the development of drugs to sleeping sickness disease. Hence, I warmly support the publication in LSA. I would like to request some minor changes to ensure the accuracy and readability of the manuscript:

1. Figure 1A - I suggest that the authors will increase the font size of the labels.
2. Figure 2A - I suggest to discuss all mutants in the text or alternatively remove the mutants that are not discussed.
3. Figure 3A - I suggest to write "(Glycosomes)" below the word "Aldolase".
4. Figure 3A, B - please discuss the non glycosomal localization of Tb Pex3.
5. Figure 3B - I suggest writing near each protein the organelle that it marks. For example: LiPDH (Mitochondria)
6. Figure 3B - the title of the x axis is missing.
7. Figure 4D - It is not clear how the authors know that the circular structures are glycosomes.
8. Table 1- It should be mentioned that the table presents the quantifications of the EMs.

9. Results, page 4 - Please add a reference to the Blast software.
10. Results, page 5 - please define the amino acids names when first mentioned.
11. Results, page 6 - It is not clear how the yeast two hybrid works if the transmembranal domain of Pex3 exists. How is Pex3 localized to the nucleus?
12. Results, page 6 - the pull-down assay (since it was not done on purified proteins) does not necessarily show a direct interaction between Pex3 and Pex19. Please rephrase.
13. Results, page 6 - It is not clear why the name of the Tb Pex3 is not consistent.
14. Figure legends, page 10 - The similarity rules are not clear.
15. Figure legends, page 12 - Please mention that the quantification of 4D is found in Table 1.
16. Figure legends, page 12, Figure 4E - error bars are currently presented for one data point - please add them for all others.
17. Materials and Methods - I suggest to move the information regarding the RNAi construct to the "RNAi and semi-quantitative RT-PCR" section and re-name the section about the Tb Pex3 cloning to: "Tb Pex3 cloning"

Reviewer #3 (Comments to the Authors (Required)):

PEX3 is central to biogenesis in peroxisomes and other microbodies. However, although trKineoplastids have microbodies called glycosomes, with many of the known components of the peroxisome biogenesis machinery, PEX3 has hitherto been elusive. IN this paper the authors identify it initially using HHpred, then confirm its likely function. They show that the protein interacts with PEX19 in a 2-hybrid experiment, and that some, but not other, conserved residues are required for the interaction. They also show that it is essential for glycosome integrity and for parasite growth.

Overall the information in this paper is really useful. It is however marred by repeated unfounded claims that Pex3 is a good drug target. This should all be removed because protein-protein interactions are usually not easy to inhibit selectively with small molecules, and there is no evidence that the Pex3-Pex19 interaction is an exception. The long lag between Pex3 loss and cell death also suggests that it is a poor target. The paper is absolutely fine without the claims about application to drug development.

Minor changes

Just one request for data. The authors recombinant protein, so please measure the sensitivity of the antibody, and the number of Pex3 molecules per cell. Then work out how long it takes for the pex3 to get diluted out after the RNAi. How low does it have to be for the cells to start to (a) lose glycosome integrity and (b) die?

1. Fig 1A - At present the selection of organisms is rather biased, there are four opisthokonts (three animals and yeast), two very closely related Excavata, one organism from Archaeplastida, and no representatives from SAR or amoebzoa. Would it be possible to change the figure, maybe removing Dr and Dm, putting Sc below Hs, and adding one representative each from the two missing supergroups? (Pex3 is expected to be absent from several Alveolata, I don't know about Stramenopiles or Rhizozoa.) This would be really useful because it would identify residues that are truly conserved throughout eukaryotic evolution.

2. Please remove all the claims that PEX3 is a good target. There is no evidence that this is the case. Similarly please delete the preamble about drugs from the Introduction. It is not relevant. Explanation: The mere fact that a protein differs from its equivalent in mammals, and is essential, does not make it a possible drug target. If it did, at least 25% of all proteins encoded by the trypanosome genome would qualify (including several other Pex proteins and most components of the secretory pathway). For a trypanosome protein to be a possible drug target it must be possible to inhibit its function selectively using a fairly small, easy-to synthesise molecule. In the case of PEX3, the function to inhibit would be a protein-protein interaction - and these are notoriously difficult to target using small (<500 Da) molecules. Another good argument would be if the equivalent interaction from mammals had already been shown to be "druggable". So far as I know, it has not. Indeed, with a few exceptions, the "target-directed" approach is now largely out of favour for development of anti-infectives. Nearly all compounds that are currently under development against parasitic diseases were found in high-throughput screens against the whole organism. The pressing priority is to get a few of these compounds into development and perhaps (though not necessarily) to find out what their targets are. Finding new targets has no priority at all. The review cited is nearly 20 years old. I suggest that the authors read something more recent, such as:

Rycker, M., B. Baragaña, S. Duce & I. Gilbert, (2018) Challenges and recent progress in drug discovery for tropical diseases. *Nature* 559: 498-506.

Barrett, M.P. & S.L. Croft, (2012) Management of trypanosomiasis and leishmaniasis. *Br Med Bull* 104: 175-196.

Giordani, F., L.J. Morrison, T.G. Rowan, D.E.K. HP & M.P. Barrett, (2016) The animal trypanosomiasis and their chemotherapy: a review. *Parasitology* 143: 1862-1889.

The review below - although it concentrates on targets - also notes that for anti-infectives, the phenotypic approach has much higher success:

Eder, J., R. Sedrani & C. Wiesmann, (2014) The discovery of first-in-class drugs: origins and evolution. *Nat Rev Drug Discov* 13: 577-587.

If the authors really think this might be a target they could test the compounds in the MMV "pathogen box" in the Pex3-Pex19 yeast 2-hybrid assay. At least 70 of those compounds kill trypanosomes.

For information see <https://www.mmv.org/mmv-open/pathogen-box/about-pathogen-box>

However I do not believe that authors are really interested in this because they have not even tested whether trypanosome Pex3 can interact with human Pex19.

Finally, DNDi and GALVmed are looking for drugs that can be given in as short a course as possible. They do not want something that takes 6 days to kill the parasite - which would mean either a large number of daily treatments, or a drug with an exceptional half-life.

3. Please supply the tritrypDB database numbers for all kinetoplastid genes mentioned - that would be Tb927.11.10260 for *T. brucei* 927 - and the homologues for other kinetoplastids (easily found at tritrypdb.org). Check whether the identified homologues are syntenic or not (also easily seen from the gene page). Also make sure that the genes are annotated. If the paper is returned for me to review I will check that this has been done.

4. RNAi gave a rather mild growth defect in a high-throughput analysis - see the gene page and Alsford, S., D. Turner, S. Obado, A. Sanchez-Flores, L. Glover, M. Berriman, C. Hertz-Fowler & D. Horn, (2011) High throughput phenotyping using parallel sequencing of RNA interference targets in the African trypanosome. *Genome Res.* 21: 915-924.

High throughput RNAi is often not correct. Especially in this paper, there is a very interesting lag before the effect is seen - which is why the screening result was wrong. It is worth pointing this out.

Could the authors speculate why there was such a long lag? Is there a threshold of protein needed in order to see an effect?

5. "complete lethality of both trypanosomal PCF and BSF cells". A cell cannot be lethal. How about "death" instead of "complete lethality"?

6. Furuya et al is not really the original reference for RNAi - it would be better to cite the reference for the plasmids used.

7. What is "numerical density" ? "Numbers" would be better.

Reviewer #1 (Comments to the Authors (Required)):

Banerjee et al. have identified Pex3 in trypanosome species. Pex3 is known to orchestrate early steps in peroxisome biogenesis, but the trypanosome ortholog has never been identified. The authors identified the candidate using HHPred. They showed that it localizes to glycosomes (the trypanosomal peroxisome analog) and that it binds to the peroxisomal chaperone Pex19, known to bind to Pex3 in other eukaryotes. They constructed strains in two species in which Pex3 was knocked down. In both strains glycosomal matrix proteins failed to localize, cells had few and aberrant glycosomes, and they failed to grow.

While there is no new insights here into peroxisomal assembly, the paper shows that this important protein and its function is conserved in trypanosoma. As cells require Pex3 for survival, Pex3 may be an important target for new drug development.

Data are straightforward and convincing. I have one significant (#1) and one minor (#2) comment:

(1) Some glycosomes are seen in the knockdown on day 3 of treatment, which is not expected in the absence of Pex3. Do you think that Pex3 is redundant or that some Pex3 protein still exists and functions? Can't the half-life of Pex3 in the knock-down be easily ascertained with the HA-tagged protein that the authors already generated? Are the number of glycosomes further reduced at day 6 or 7 before the cells die?

We have addressed the question of the kinetics of knock down using RT-PCR of *TbPex3* transcript (see new Fig. S1). Please note that it is not possible to do both RNAi and expression of *TbPex3*-HA because the plasmid constructs for each uses the same drug selection (phleomycin).

We observe a rapid reduction in the abundance of *Pex3* transcript upon RNAi treatment (Fig. S1, A). This reduction continues over time. After 8 h of RNAi treatment, there is little *Pex3* transcript remaining. Concomitantly, visible defects in glycosome morphology appear early after the start of *TbPex3*-RNAi treatment and exacerbate with continued RNAi treatment (Fig. S1, B). These defects can be characterized as release of the glycosomal matrix protein aldolase to the cytosol and the fragmentation of glycosomes.

We can only speculate as to why it takes 6 days for the effects of *TbPex3*-RNAi knock down to kill all BSF cells (see p. 9 of the revised manuscript). It is noteworthy that an independent study by Alsford and colleagues in 2011 also reported no significant loss of fitness of BSF cells after 3 days of *TbPex3*-RNAi treatment, but a significant loss of fitness after 6 days of treatment. The Alsford et al. manuscript and its findings are now referenced on p. 9 of the revised manuscript.

(2) Page 5 states that ". . . position 104 is therefore sufficient for the binding of HsPex3". Do the data only show that position 104 is necessary (but not necessarily sufficient)?

The Reviewer is correct, and we have changed 'sufficient' to 'necessary'.

Reviewer #2 (Comments to the Authors (Required)):

Glycosomes are peroxisome-like organelles that are essential for the viability of *Trypanosoma brucei* (Tb). Although several proteins that contribute to the biogenesis and function of glycosomes were identified in the past, Pex3, a key protein that exists in peroxisomes in all eukaryotes, was not identified. In their manuscript, Banerjee et al. identified the Tb Pex3 protein by using the HHpred tool. They show that the putative Tb Pex3 is similar to well-studied Pex3 proteins in other organisms, that Tb Pex3 binds to Tb Pex19 (as do all other studied Pex3 proteins to date), and that reducing the amounts of Tb Pex3 drastically affects the formation of glycosomes. Moreover, they show that Tb Pex3 is essential for the viability of the parasite and hence suggest that Tb Pex3 is a potential drug target.

The data seem solid, the experiments well performed, the manuscript well written, the scientific story is well presented. Importantly, the identification of the Tb Pex3 is an important piece in the cell biology of this parasite and may contribute to the development of drugs to sleeping sickness disease. Hence, I warmly support the publication in LSA. I would like to request some minor changes to ensure the accuracy and readability of the manuscript:

1. Figure 1A - I suggest that the authors will increase the font size of the labels.

Done.

2. Figure 2A - I suggest to discuss all mutants in the text or alternatively remove the mutants that are not discussed.

All mutants are now discussed (p. 6 of the revised manuscript).

3. Figure 3A - I suggest to write "(Glycosomes)" below the word "Aldolase". Figure 3B - I suggest writing near each protein the organelle that it marks. For example: LiPDH (Mitochondria)

Done.

4. Figure 3A, B - please discuss the non glycosomal localization of Tb Pex3.

We now discuss the non-glycosomal localization of TbPex3 on p. 7 of the revised manuscript. The non-glycosomal localization of TbPex3 is likely due to its trafficking through the ER, as has been shown for Pex3 proteins in other organisms. We also note the partial cofractionation of TbPex3 with the ER protein, BiP (Fig. 3, B)

6. Figure 3B - the title of the x axis is missing.

We thank the Reviewer for pointing this out. The x-axis has now been labeled.

7. Figure 4D - It is not clear how the authors know that the circular structures are glycosomes.

Glycosomes have previously been characterized as spherical organelles surrounded by a single unit membrane and containing a somewhat granular matrix. We now reference two publications that provide transmission electron microscopic images of glycosomes (Banerjee et al., 2005; de Souza, 2008; p. 8 of the revised manuscript).

8. Table 1- It should be mentioned that the table presents the quantifications of the EMs.

Done.

9. Results, page 4 - Please add a reference to the Blast software.

Done (Altschul et al., 1990; p. 4 of the revised manuscript).

10. Results, page 5 - please define the amino acids names when first mentioned.

Done (pp. 5 and 6 of the revised manuscript).

11. Results, page 6 - It is not clear how the yeast two hybrid works if the transmembranal domain of Pex3 exists. How is Pex3 localized to the nucleus?

Hydrophobic structure predictions indicate a clear transmembrane region at the N-terminus of *Saccharomyces cerevisiae* Pex3. We removed amino acids 1-50 of ScPex3 encompassing this region for yeast 2-hybrid analysis.

TbPex3 does not have a clearly defined transmembrane region. Analysis was therefore done on the entire molecule.

12. Results, page 6 - the pull-down assay (since it was not done on purified proteins) does not necessarily show a direct interaction between Pex3 and Pex19. Please rephrase.

We have removed 'direct'.

13. Results, page 6 - It is not clear why the name of the Tb Pex3 is not consistent.

Different nomenclatures are found in the field. We have tried to clarify the confusion by providing the protein accession number and corresponding encoding gene designation for all proteins.

14. Figure legends, page 10 - The similarity rules are not clear.

The similarity rules have been clarified in the revised legend to Figure 1.

15. Figure legends, page 12 - Please mention that the quantification of 4D is found in Table 1.

Done (p. 13 of the revised manuscript).

16. Figure legends, page 12, Figure 4E - error bars are currently presented for one data point - please add them for all others.

Error bars were calculated for all data points (measurements were done in triplicate); however, the errors for all time points except PCF, 8 days are so small as to not exceed the width of the graph symbols themselves.

17. Materials and Methods - I suggest to move the information regarding the RNAi construct to the "RNAi and semi-quantitative RT-PCR" section and re-name the section about the Tb Pex3 cloning to: "Tb Pex3 cloning"

Done.

Reviewer #3 (Comments to the Authors (Required)):

PEX3 is central to biogenesis in peroxisomes and other microbodies. However, although trKineoplastids have microbodies called glycosomes, with many of the known components of the peroxisome biogenesis machinery, PEX3 has hitherto been elusive. IN this paper the authors identify it initially using HHpred, then confirm its likely function. They show that the protein interacts with PEX19 in a 2-hybrid experiment, and that some, but not other, conserved residues are required for the interaction. They also show that it is essential for glycosome integrity and for parasite growth.

Overall the information in this paper is really useful. It is however marred by repeated unfounded claims that Pex3 is a good drug target. This should all be removed because protein-protein interactions are usually not easy to inhibit selectively with small molecules, and there is no evidence that the Pex3-Pex19 interaction is an exception. The long lag between Pex3 loss and cell death also suggests that it is a poor target. The paper is absolutely fine without the claims about application to drug development.

All text referring to drug development has been removed from the revised manuscript.

Minor changes

Just one request for data. The authors recombinant protein, so please measure the sensitivity of the antibody, and the number of Pex3 molecules per cell. Then work out how long it takes for the pex3 to get diluted out after the RNAi. How low does it have to be for the cells to start to (a) lose glycosome integrity and (b) die?

As outlined in our response to Reviewer #1 (see above), we are unable to simultaneously perform *TbPex3*-RNAi and measure the quantity of recombinant TbPex3-HA present in cells because the plasmids for *TbPex3*-RNAi and expression of TbPex3-HA use the same drug selection, phleomycin. To provide some information on the kinetics of knock down of *TbPex3* transcript, we measured the abundance of *TbPex3* transcript over time and compared the reduction in *TbPex3* transcript levels with glycosome integrity over time (see new Fig. S1).

1. Fig 1A - At present the selection of organisms is rather biased, there are four opisthokonts (three animals and yeast), two very closely related Excavata, one organism from Archaeplastida, and no representatives from SAR or amoebzoa. Would it be possible to change the figure, maybe removing Dr and Dm, putting Sc below Hs, and adding one representative each from the two missing supergroups? (Pex3 is expected to be absent from several Alveolata, I don't know about Stramenopiles or Rhizozoa.) This would be really useful

because it would identify residues that are truly conserved throughout eukaryotic evolution.

The figure has been modified according to the Reviewer's suggestions. We removed the fish and fly sequences and replaced them with sequences from a representative of the amoebozoia (*Planoprotostelium fungivorum*) and a representative of the SAR supergroup (a stramenopile, *Nannochloropsis gaditana*). Table S1 has been modified accordingly. It is noteworthy that residues of Pex3 that are necessary for interaction with Pex19 in human, plants and yeast are conserved in the Pex3 proteins from these more evolutionarily distant organisms.

2. Please remove all the claims that PEX3 is a good target. There is no evidence that this is the case. Similarly please delete the preamble about drugs from the Introduction. It is not relevant. Explanation: The mere fact that a protein differs from its equivalent in mammals, and is essential, does not make it a possible drug target. If it did, at least 25% of all proteins encoded by the trypanosome genome would qualify (including several other Pex proteins and most components of the secretory pathway). For a trypanosome protein to be a possible drug target it must be possible to inhibit its function selectively using a fairly small, easy-to-synthesise molecule. In the case of PEX3, the function to inhibit would be a protein-protein interaction - and these are notoriously difficult to target using small (<500 Da) molecules. Another good argument would be if the equivalent interaction from mammals had already been shown to be "druggable". So far as I know, it has not. Indeed, with a few exceptions, the "target-directed" approach is now largely out of favour for development of anti-infectives. Nearly all compounds that are currently under development against parasitic diseases were found in high-throughput screens against the whole organism. The pressing priority is to get a few of these compounds into development and perhaps (though not necessarily) to find out what their targets are. Finding new targets has no priority at all.

The review cited is nearly 20 years old. I suggest that the authors read something more recent, such as:

Rycker, M., B. Baragaña, S. Duce & I. Gilbert, (2018) Challenges and recent progress in drug discovery for tropical diseases. *Nature* 559: 498-506.

Barrett, M.P. & S.L. Croft, (2012) Management of trypanosomiasis and leishmaniasis. *Br Med Bull* 104: 175-196.

Giordani, F., L.J. Morrison, T.G. Rowan, D.E.K. HP & M.P. Barrett, (2016) The animal trypanosomiasis and their chemotherapy: a review. *Parasitology* 143: 1862-1889.

The review below - although it concentrates on targets - also notes that for anti-infectives, the phenotypic approach has much higher success:

Eder, J., R. Sedrani & C. Wiesmann, (2014) The discovery of first-in-class drugs: origins and evolution. *Nat Rev Drug Discov* 13: 577-587.

If the authors really think this might be a target they could test the compounds in the MMV "pathogen box" in the Pex3-Pex19 yeast 2-hybrid assay. At least 70 of those compounds kill trypanosomes.

For information see <https://www.mmv.org/mmv-open/pathogen-box/about-pathogen-box>

However I do not believe that authors are really interested in this because they have not even tested whether trypanosome Pex3 can interact with human Pex19.

Finally, DNDi and GALVmed are looking for drugs that can be given in as short a course as possible. They do not want something that takes 6 days to kill the parasite - which would mean either a large number of daily treatments, or a drug with an exceptional half-life.

All mention of TbPex3 being a good drug target has been removed from the revised manuscript.

We have now cited De Rycker et al., 2018 and Giordani et al., 2016 in the revised manuscript.

3. Please supply the tritrypDB database numbers for all kinetoplastid genes mentioned - that would be Tb927.11.10260 for *T. brucei* 927 - and the homologues for other kinetoplastids (easily found at tritrypdb.org). Check whether the identified homologues are syntenic or not (also easily seen from the gene page). Also make sure that the genes are annotated. If the paper is returned for me to review I will check that this has been done.

This has been done and is now presented as a new Table S2.

4. RNAi gave a rather mild growth defect in a high-throughput analysis - see the gene page and Alford, S., D. Turner, S. Obado, A. Sanchez-Flores, L. Glover, M. Berriman, C. Hertz-Fowler & D. Horn, (2011) High throughput phenotyping using parallel sequencing of RNA interference targets in the African trypanosome. *Genome Res.* 21: 915-924.
High throughput RNAi is often not correct. Especially in this paper, there is a very interesting lag before the effect is seen - which is why the screening result was wrong. It is worth pointing this out. Could the authors speculate why there was such a long lag? Is there a threshold of protein needed in order to see an effect?

The Alford et al. paper reports no loss of fitness through day 3 of *TbPex3*-RNAi treatment of BSF cells, but a loss of fitness by day 6 of RNAi treatment (see their Table S1 and explanations contained therein). The effects on BSF cell viability of *TbPex3*-RNAi knock down that we observed resemble those observed Alford et al. We analyzed the kinetics of RNAi knock down of *TbPex3* transcript and the effects of this knock down over time on glycosome integrity (see new Fig. S1). We can only speculate as to why it takes 6 days for the effects of *TbPex3*-RNAi knock down to kill all BSF cells (see p. 9 of the revised manuscript).

5. "complete lethality of both trypanosomal PCF and BSF cells". A cell cannot be lethal. How about "death" instead of "complete lethality"?

'death' has been substituted for 'complete lethality'.

6. Furuya et al is not really the original reference for RNAi - it would be better to cite the reference for the plasmids used.

Done.

7. What is "numerical density" ? "Numbers" would be better.

Changed in a new Table 1, as suggested above.

i

July 16, 2019

RE: Life Science Alliance Manuscript #LSA-2019-00421-TR

Dr. Richard A. Rachubinski
University of Alberta
Cell Biology
Medical Sciences Bldg. 5-14
Medical Sciences Building 5-14
Edmonton, Alberta T6G 2H7
Canada

Dear Dr. Rachubinski,

Thank you for submitting your revised manuscript entitled "The early acting glycosome biogenic protein Pex3 is essential for trypanosome viability". As you will see, the reviewers appreciate the introduced changes and we would thus be happy to publish your paper in Life Science Alliance.

For the final version, please check and make sure one more time to indicate the number of replicates for the various experiments performed.
Please fill in the electronic license to publish form in our submission system.

A. FINAL FILES:

-- Summary blurb (enter in submission system): A short text summarizing in a single sentence the study (max. 200 characters including spaces). This text is used in conjunction with the titles of papers, hence should be informative and complementary to the title. It should describe the context

and significance of the findings for a general readership; it should be written in the present tense and refer to the work in the third person. Author names should not be mentioned.

B. MANUSCRIPT ORGANIZATION AND FORMATTING:

Sincerely,

Andrea Leibfried, PhD
Executive Editor
Life Science Alliance
Meyerohofstr. 1
69117 Heidelberg, Germany
t +49 6221 8891 502
e a.leibfried@life-science-alliance.org
www.life-science-alliance.org

Reviewer #1 (Comments to the Authors (Required)):

I had only minor comments. They have all been well addressed. I don't see that any significant comments of the other reviewers have not been addressed as well. No new critical issues.

Reviewer #3 (Comments to the Authors (Required)):

The authors have responded to my requests adequately and the new alignment does, I hope give more useful information than the original.

July 17, 2019

RE: Life Science Alliance Manuscript #LSA-2019-00421-TRR

Dr. Richard A. Rachubinski
University of Alberta
Cell Biology
Medical Sciences Bldg. 5-14
Medical Sciences Building 5-14
Edmonton, Alberta T6G 2H7
Canada

Dear Dr. Rachubinski,

Thank you for submitting your Research Article entitled "The early acting glycosome biogenic protein Pex3 is essential for trypanosome viability". It is a pleasure to let you know that your manuscript is now accepted for publication in Life Science Alliance. Congratulations on this interesting work.

*****IMPORTANT:** If you will be unreachable at any time, please provide us with the email address of an alternate author. Failure to respond to routine queries may lead to unavoidable delays in publication.*******

DISTRIBUTION OF MATERIALS:

Again, congratulations on a very nice paper. I hope you found the review process to be constructive and are pleased with how the manuscript was handled editorially. We look forward to future exciting

submissions from your lab.

Sincerely,
